# Effects of Glycine Supplementation in Drinking Water on the Growth Performance, Intestinal Development, and Genes Expression in the Jejunum of Chicks

**DOI:** 10.3390/ani13193109

**Published:** 2023-10-05

**Authors:** Xiaotong Zheng, Yinku Xie, Ziwei Chen, Jiaheng He, Jianfei Chen

**Affiliations:** 1School of Biotechnology, Jiangsu University of Science and Technology, Zhenjiang 212100, China; xiaotzheng@just.edu.cn (X.Z.); xyk20030501@163.com (Y.X.); chenziwei20222022@163.com (Z.C.); 15656083058@163.com (J.H.); 2Key Laboratory of Silkworm and Mulberry Genetic Improvement, Ministry of Agriculture and Rural Affairs, Chinese Academy of Agricultural Sciences, Zhenjiang 212100, China

**Keywords:** chick, glycine, growth performance, intestinal health

## Abstract

**Simple Summary:**

The objective of this study was to investigate the effects of glycine supplementation in drinking water on the growth performance and intestinal development of newborn chicks. Parameters such as body weight, organ indexes, intestinal villus development, and the expression of genes in the intestine were assessed. The findings revealed that glycine supplementation in drinking water resulted in an increase in the average daily gain (ADG) of chicks aged 7 to 14 days. An increase in jejunum villus height and the ratio of villus height to crypt depth was also observed, which could be considered the possible factor contributing to the enhanced ADG. Furthermore, glycine supplementation in drinking water resulted in changes in the expression of a few genes related to mucosal immune barrier, cellular capacity against oxidation, and amino acid transporter in jejunum. These findings indicate that glycine supplementation in drinking water has the potential to enhance growth efficiency and promote intestinal health in chicks.

**Abstract:**

Glycine, the most basic amino acid found in nature, is considered an essential amino acid for chicks. However, the precise understanding of high concentrations of glycine’s significance in promoting the growth performance of chicks, as well as its impact on intestinal development, re-mains limited. Consequently, the objective of this study was to investigate the effects of glycine supplementation in drinking water on growth performance, intestine morphology, and development in newly hatched chicks. In this study, 200 newly born chicks were selected and pro-vided with a supplementation of 0.5%, 1%, and 2% glycine in their drinking water during their first week of life. The results revealed that glycine supplementation in drinking water could significantly increase the average daily gain of chicks from days 7 to 14. Furthermore, a significant difference was observed between the group supplemented with 1% glycine and the control group. Concurrently, this glycine supplementation increased the villus height and the ratio of the villus height to crypt depth in jejunum on both day 7 and day 14. Glycine supplementation in drinking water significantly affected the mRNA expression level of the *ZO-1*, *GCLM*, and *rBAT* genes in jejunum, which may have certain effects on the mucosal immune defense, cellular antioxidant stress capacity, and amino acid absorption. Overall, the findings of this study indicate that glycine supplementation in drinking water can enhance the growth performance of chicks and promote their intestine development.

## 1. Introduction

Glycine is the smallest and simplest amino acid that can be synthesized by mammals such as humans, pigs, and mice [1]. However, chicks are unable to produce enough glycine to meet their own needs, thus rendering it a conditionally essential amino acid [2]. Glycine serves as a significant substrate for the synthesis of purines, the neurotransmitter glutathione, and other biomolecules. Additionally, it is involved in the synthesis of cellular proteins; it is an amino acid required for the development of newborns [3,4]. Moreover, glycine plays crucial roles in antioxidant mechanisms [5], cellular protection [6], and immune regulation [7]. Therefore, a certain proportion of glycine should be added to base diets to ensure chicks’ normal operation of life activities.

Chicks experience a shift from relying on internal nutrition (yolk sac) to external nutrition (feed) by the time they reach 7 days old, a critical stage in their development [8]. During this stage, the intestinal tract of young chicks remains underdeveloped, rendering it highly susceptible to external stimuli, which may result in infection and inflammation [9]. Inflammation can cause damage and impairment of the intestinal mucosa, thereby adversely impacting productivity and overall health [10]. A robust intestinal tract not only assumes a pivotal role in the efficient digestion and assimilation of nutrients, but also prevents the invasion and dissemination of pathogens [11]. Hence, the maintenance of intestinal health and optimal development of chicks via nutritional interventions is crucial for promoting their subsequent growth and overall well-being.

Elevated levels of amino acids have been found to mitigate intestinal inflammation and sustain intestinal health [12]. Studies have proved that dietary glycine supplementation prevents heat-stress-induced impairment of antioxidant status and intestinal barrier function in broilers [13]. The dietary supplementation of glycine can significantly alleviate intestinal injury and protect intestinal cells against oxidative stress in piglets [14,15]. Furthermore, the oral administration of glycine following weaning has shown potential in enhancing the integrity of the intestinal mucosal barrier by augmenting the expression of tight junction proteins [16,17]. Research has demonstrated that adding glycine to the diet of rats can effectively mitigate the incidence of diarrhea induced by enteritis [18]. Dietary supplementation with glycine also protected the structural integrity and thickness of the mucosal epithelium of the intestinal in rats exposed to abdominal radiation [19]. Currently, investigations on glycine primarily concentrate on the supplementation of low-protein diets in chicks. It is possible to substantially decrease dietary crude protein (CP) in poultry below the minimum CP requirement by adjusting the dietary glycine equivalents [20,21]. However, there is a lack of research regarding the effects of glycine supplementation in drinking water on the growth performance and intestinal development of chicks.

Investigating the early nutritional regulation of chicks, particularly the impact of nutritional factors on body growth and organ development, especially the intestinal tract, is of utmost importance [22]. We hypothesized that glycine may play a role in enhancing intestinal health in chicks. We identified the expression of the genes-related Kelch-like ECH-associated protein 1/nuclear factor erythroid 2-related 2 (Keap1/Nrf2) signaling pathway and tight junction protein. The Keap1/Nrf2 pathway is the principal protective response to oxidative and electrophilic stresses. Tight junction proteins are crucial for the maintenance of epithelial barrier integrity. In addition, the expression of some common amino acid transporter genes was also identified to determine the transport efficiency of glycine. This study aimed to examine the effects of glycine supplementation in drinking water on the growth performance and intestinal development of chicks, specifically focusing on factors such as body weight, organ index, intestinal villi development, and the expression of genes related to the mucosal immune barrier, cellular capacity against oxidation, and the amino acid transporter, thereby optimizing the provision of appropriate nutrition and fostering the body development of chicks.

## 2. Materials and Methods

### 2.1. Animals and Treatments

A total of 200 newly hatched female Hyland brown chickens were procured from a commercial hatchery located in Nanjing, Jiangsu province, China. Two hundred chicks with comparable weight were selected and randomly assigned to four groups, with each group consisting of five replicate cages and ten chicks per cage. Throughout the first two weeks, all groups were provided with a commercial corn–soybean meal diet. The base diet was formulated based on the National Research Council (1994) [23], with 21% crude protein, 1.02% calcium, 0.5% total phosphorus, 6% crude fiber, 1.16% digestible lysine, 0.84% digestible methionine + cystine, 0.82% digestible glycine, and 0.92 digestible serine. The control group received regular drinking water for two weeks (days 1–14), while the remaining three groups were administered regular drinking water supplemented with 0.5%, 1.0%, and 2.0% glycine (ABCONE, Shanghai, China) during the initial week (days 1–7), and regular drinking water in the second week (days 8–14). Several previous studies led to the choice of the glycine concentration [13,24]. The broilers were housed in stainless steel cages with continuous light and unrestricted access to feed and water.

### 2.2. Performance Measurement and Sampling

The chicks were weighed at 0, 7, and 14 days of age to calculate average daily gain (ADG). At the ages of 7 and 14 days, ten chicks from each treatment group (2 chicks per replicate) were euthanized using cervical dislocation. The chickens fasted for 12 h before body weight measurement and euthanasia, and their gastrointestinal tracts were emptied prior to weighing them. The heart, liver, spleens, lungs, kidneys, gizzard, duodenum, jejunum, and ileum were dissected and individually weighed. The organ index was then calculated by determining the percentage of organ weight in relation to body weight (BW). The middle portion of jejunum was collected for histology and mRNA expression analysis. Jejunum samples measuring approximately 1 cm were isolated, washed with phosphate buffers (Solarbio, Beijing, China), and fixed in 4% paraformaldehyde (Beyotime, Shanghai, China) for morphological analysis. Additionally, the jejunum samples were frozen in liquid nitrogen and subsequently stored at −80 °C for further analysis.

### 2.3. Intestinal Morphology Determination

Intestinal samples were fixed in 4% paraformaldehyde (Beyotime, Shanghai, China), dehydrated in ethanol (Macklin, Shanghai, China) and xylene alcohol (Macklin, Shanghai, China), and subsequently washed and embedded in paraffin wax (Leyan, Shanghai, China). The paraffin-embedded samples were then sectioned into 5 μm slices using a Leica RM2135 microtome (Leica, Wetzlar, Germany) and placed onto glass slides. Dewaxing of the paraffin was achieved using xylene, followed by rehydration in a series of graded alcohol solutions. The sections were stained with hematoxylin–eosin staining and examined under an Olympus microscope (IX73, magnification 40×). The villus height and crypt depth were measured using OlyVIA software, and the ratio of villus height to crypt depth (V/C) was subsequently calculated.

### 2.4. Quantitative Real-Time PCR

Jejunum tissues were subjected to RNA extraction using TRIzol (Takara, Dalian, China), followed by reverse transcription of 1 μg total RNA into cDNA using the PrimeScriptTM First Strand cDNA Synthesis Kit (Takara, Dalian, China). A quantitative real-time polymerase chain reaction (qRT-PCR) was conducted using the CFX96TM Real-Time System (Bio-Rad, Hercules, CA, USA), with three technical replicates per sample. The reaction volume was 20 μL, which contained 1 μL cDNA, 10 μL qPCR SYBR Green Master Mix (Vazyme, Nanjing, China), 1 μL of each of the forward and reverse primers (10 μM), and 7 μL deionized water. The qPCR amplification procedure was as follows: 95 °C for 15 min, 40 cycles of 95 °C for 10 s, 58 °C for 20 s, 72 °C for 30 s, and an extension for 10 min at 72 °C. The housekeeping gene adaptor-related protein complex 2 mu 1 subunit (*AP2M1*) and RNA polymerase II subunit B (*POLR2B*) were used for normalization [25]. The primer sequences utilized are provided in Table 1.

### 2.5. Statistical Analysis

SPSS 26.0 software was used to conduct a one-way ANOVA, and Duncan’s multiple range test was used to detect significant differences between individual means when the treatment effect was significant. *p* < 0.05 was considered statistically significant. The gene relative expression was determined using the 2^−ΔΔCT^ method. The genomic mean results of 2 housekeeping genes were considered as normalizers to attain the gene expression profile in this study. Statistical analysis was performed using paired Student’s *t*-tests in the SPSS 26 software. The results were expressed as the mean and standard error of the mean.

## 3. Results

### 3.1. Body Weight and Average Daily Gain

The groups did not exhibit significant differences in BW on days 0, 7, and 14 (Table 2). There was a tendency for ADG to increase with increasing concentrations of glycine from 0 to 7 days and 7 to 14 days. Glycine supplementation in drinking water had no effect on the ADG of chicks from day 0 to 7, but it had a significant impact on the ADG of chicks from day 7 to 14 (*p* < 0.05). During this period, the ADG was significantly higher in the chicks treated with 1% glycine compared to the control group (*p* < 0.05). Although there was no significant difference in ADG between the 2% glycine treatment and the control group, the ADG increased by 11.00%.

### 3.2. Organ Development

The impact of glycine supplementation in drinking water on the organ index of chicks is shown in Figure 1. The findings revealed that, on day 7, the liver organ index in the group supplemented with 0.5% glycine was significantly elevated compared to the 1% and 2% groups (*p* < 0.05), and there was no difference between the three supplementation groups and the control group. For the ileum organ index, there was only a statistical difference between the 0.5% glycine supplementation group and the control group. Conversely, no significant alterations were observed in the remaining organ indexes. Furthermore, at 14 days, following the discontinuation of glycine supplementation for 7 days, the heart organ indexes in the initial 0.5% glycine supplementation group exhibited a significant decrease relative to the other groups, and no significant alterations were observed in the remaining organ indexes.

### 3.3. Intestinal Morphology

The analysis of the intestinal morphology indicates that glycine supplementation in drinking water had a significant impact on the villus height and V/C ratio of the jejunum on days 7 and 14 (*p* < 0.05) and had no significant effect on crypt depth (Figure 2). Specifically, on day 7, the villus height of the jejunum increased by 21.03%, 30.78%, and 27.08% in the 0.5%, 1%, and 2% glycine supplementation groups, respectively. And there was a notable difference between the 1% glycine supplementation group and the control group. In addition, all three groups exhibited a significant increase in the V/C ratio compared to the control group (*p* < 0.05). On the 14th day, the villus height of the jejunum in the 1% and 2% groups exhibited statistically significant increases of 22.40% and 22.56%, respectively. Additionally, the V/C ratio in both groups showed a significant increase when compared to the control group (*p* < 0.05). However, no significant difference in the villus height and V/C ratio was observed in the 0.5% glycine supplementation group compared to the control group, potentially due to the lower glycine supplementation.

### 3.4. Expression of Tight Junction Protein-Related, Keap1/Nrf2 Signaling Pathway-Related, and Amino Acid Transporter Genes in Jejunum

The expression of tight junction protein-related genes in the jejunum from the 7th day is documented in Table 3. Glycine supplementation in the drinking water resulted in an up-regulation of the tight junction protein 1 (*ZO-1*) gene, with a significant difference observed between the 2% glycine supplementation group and the control group. Glycine supplementation in drinking water had no effect on the expression of other tight junction protein-related genes, including *Occludin*, *ZO-2*, *Claudin1*, *Claudin2*, and *Claudin3*. The mRNA expression data of the Keap1/Nrf2 signaling pathway in the jejunum is presented in Table 4. In comparison to the control group, the mRNA expression of glutamate–cysteine ligase modifier subunit (*GCLM*) decreased in the glycine-supplemented group, with a significant difference observed in the 2% glycine group. The expression levels of superoxide dismutase 1 (*SOD1*), glutathione–disulfide reductase (*GSR*), and glutathione peroxidase (*GPX*) exhibited a decrease, although the differences were not statistically significant when compared to the control group. Glycine supplementation in drinking water increased the expression of the *Keap1* gene in all groups, and the expression of the *NQO1* gene first increased and then decreased with the increase in the glycine concentration. However, the difference was not significant compared with the control group. The effects of glycine supplementation in drinking water on the mRNA expression of amino acid transporter genes in jejunum are shown in Table 5. Specifically, the mRNA expression of solute carrier family 3 members 1 (*rBAT*) experienced a decrease in the jejunum, and the 1% and 2% groups showed a statistically significant difference compared to the control group (*p* < 0.05). However, the expression of the other three amino acid transporter genes did not show significant changes.

## 4. Discussion

The health of newly hatched chicks holds an important economic implication as it directly impacts both their survival rate and subsequent growth performance. Glycine is an essential functional amino acid for chicks, and insufficient glycine levels detrimentally affect the growth performance of chicks [26]. Consequently, it is necessary to supplement the base diet of chicks with a specific proportion of glycine [27]. In previous studies, glycine was added to a base diet, and an effect on the growth performance was observed [13,26,27]. In this study, we try to increase the amount of glycine in the gastrointestinal tract by adding glycine to drinking water, and then we assessed its effect on the growth performance and intestinal development of chicks.

Our study has revealed a significant enhancement of the ADG of chicks aged 7 to 14 days upon 1% glycine supplementation in drinking water. We postulate that glycine potentially exerts influences on the stimulate appetite and feed consumption in chicks. This is attributed to the sweet taste of glycine, which may attract chicks to eat it, as well as promoting the hypothalamic pituitary gland to secrete hormones such as the growth hormone and prolactin [18]. Glycine also functions as a neurotransmitter within the central nervous system, playing a regulatory role in behavior and food intake [28]. Interestingly, glycine supplementation did not lead to a significant increase in ADG at 0–7 days of age. This may be because glycine supplementation enhances white-fat loss and improves sensitivity to insulin in animals [29]. The ADG on 7–14 days increased significantly, which may be due to the glycine supplementation which occurred on days 0–7 to promote intestinal villus development and absorption capacity. Furthermore, we assessed the impact of glycine supplementation in drinking water on the organ index of chicks. Generally, organs exhibit concurrent growth with the body, albeit with distinct growth curves upon their respective functions [30]. However, most organ indexes did not show significant differences between the different groups. We believe that the organs in the glycine supplementation group underwent a better development, and the rapid development of multiple organs led to the increase in ADG.

The development of the small intestine is particularly critical during the initial week following hatching [31,32]. Serving as an organ for digestion and nutrient absorption in animals, the small intestine performs the essential functions of digesting and absorbing nutrients, minerals, and water [33]. The digestive and absorptive capacity of the small intestine depends on the integrity of its villi. Research indicates that glycine supplementation in milk replacer has the potential to enhance the villus height and the V/C value in neonatal piglets [24]. This study revealed that glycine supplementation in drinking water can stimulate the growth of intestinal villi, primarily evidenced by an increase in villus height and V/C value. Villus height, crypt depth, and V/C value are significant indicators of intestinal health, with the V/C value serving as a dependable indicator of intestinal development [34]. The elevation in villi height is tantamount to an expansion in the surface area of intestinal digestion and absorption, thereby enhancing the capacity for intestinal digestion and absorption [35]. Consequently, glycine supplementation in the drinking water has the potential to amplify the transportation, digestion, and absorption capacity of intestinal nutrients, thereby improving the growth performance of chicks. And this advantage effect is continuous; one week after the cessation of glycine supplementation, the intestinal villi of original supplementation groups still showed better development. This may depend on a better early developmental foundation and glycine storage.

As a crucial medium for communication between the internal and external environment, the intestine also plays an immune barrier function [36]. During the initial stages of chick development, the mechanical barrier formed by the intestinal mucosal epithelial cells and their tight junction proteins assumes a vital role in resisting pathogen invasion. Occludin, claudin, and zonula occluden stand out as pivotal members of the tight junction protein family [37]. They possess the ability to modulate the permeability of the intestinal barrier, as well as maintain and regulate the barrier function of the intestinal epithelium. Additionally, they participate in the transport of cellular materials and uphold epithelial polarity [38]. *Salmonella* infection has been observed to down-regulate the expression of *Occludin* and *ZO-1* genes in the jejunum, as well as *Occludin* and *Claudin* genes in the ileum [39]. The expression of the *ZO-1* gene in intestinal epithelial cells has been found to mitigate bacterial-induced intestinal injury [40]. Furthermore, previous research has demonstrated that the physiological levels of glycine can enhance the integrity of the intestinal epithelial barrier in piglets [16]. This protective effect is attributed to an increase in the expression and distribution of tight junction proteins, such as *claudin-7* and *ZO-3* in intestinal epithelial cells [17]. In this study, it was observed that the expression of the *ZO-1* gene in the jejunum of the glycine supplement group on day 7 was significantly higher when compared to the control group. However, the expression levels of other genes related to tight junction proteins, namely *Occludin*, *ZO-2*, *Claudin1*, *Claudin2*, and *Claudin3*, were lower in the glycine group compared to the control group. Notably, the expression levels of *ZO-2* and *Claudin1* genes were almost significantly reduced in the glycine group, with *p*-values of 0.07 and 0.06, respectively. Further investigation is required to determine if the high expression of *ZO-1* can compensate for the low expression of other tight junction proteins and subsequently enhance the integrity of the intestinal barrier. Additionally, the expression of *Occludin*, *Claudin1*, and *Claudin2* exhibited a slight increase in the low-concentration glycine supplement group, but this decreased once again in the high-concentration group. Further investigation is necessary to ascertain whether a low concentration of glycine can promote the expression of these tight junction proteins.

The *Keap1/Nrf2* pathway is a crucial antioxidant defense mechanism in the gastrointestinal tract [41] responsible for regulating the expression of various antioxidant-related genes, including *NQO1*, *GPX,* and the genes involved in glutathione synthesis [42,43,44]. NQO1, an antioxidant enzyme and flavin, is a significant antioxidant substance in the body [45]. The findings of this study indicate that the *NQO1* gene exhibited a tendency of high expression in the group supplemented with low concentrations of glycine, whereas the expression level decreased in the group supplemented with high concentrations (2%) of glycine (*p* < 0.05). These results suggest that a low concentration of glycine may elevate antioxidant stress levels, while a high concentration of glycine may induce oxidative stress. The *Keap1* gene in the glycine supplementation group showed an increasing trend in comparison to the control group (*p* > 0.05). Conversely, the expression levels of other antioxidant stress factors, including *GCLM*, *SOD1*, *GSR*, and *GPX*, all demonstrated a decreased trend, with particular emphasis on the significantly reduced level of the *GCLM* gene (*p* < 0.05). This observation suggests that glycine may elevate oxidative stress levels in the intestinal region of chicks. GCLM is a vital component of G-glutamylcysteine ligase and acts as a rate-limiting enzyme in the synthesis of glutathione [46,47]. In our study, glycine supplementation resulted in a reduction in the mRNA expression of *GCLM* in the jejunum, potentially leading to a limitation in the synthesis of glutathione. This finding aligns with a previous study conducted by Jackson et al., which observed a decrease in the erythrocyte glutathione concentration and synthesis rate following the intravenous infusion of 20 mmol/L glycine [48]. Consequently, while glycine supplementation significantly enhances the growth performance of chicks, it may also contribute to increased intestinal oxidative stress. Nevertheless, it is still worthwhile to investigate whether a suitable concentration of glycine can effectively elevate antioxidant stress levels.

The specific glycine transporter solute carrier family 6 member 9 (GLYT1) is responsible for some 30–50% of glycine uptake into intestinal epithelial cells [49]. The absorption of neutral amino acids across the luminal membrane of intestinal enterocytes is mediated by solute carrier family 6 member 19 (B^0^AT1) [50]. The rBAT protein is a carrier for cysteine, basic, and neutral amino acids [51]. Although the transport of amino acids is mediated by several specific amino acid transporters, the uptake of more than 8000 different di- and tripeptides is performed by solute carrier family 15 member 1 (PepT1) [52]. In this study, with the exception of *GLYT1*, which exhibited an upward trend in terms of expression at a glycine concentration of 0.5%, its expression level demonstrated a declining pattern under all other concentration conditions. Furthermore, other amino acid transporter genes (*PepT1*, *B^0^AT1*, and *rBAT*) displayed a tendency towards decreased expression, with *rBAT* exhibiting a notably significant decrease. From a gene expression standpoint, it has been elucidated that glycine supplementation may diminish the expression of amino acid transporters, consequently impacting the intestinal absorption of associated amino acids. However, this study also suggests that glycine supplementation yields a more favorable impact on the development of intestinal villi. This enhanced development of the intestinal villi system may serve as a compensatory mechanism for the impaired nutrient absorption resulting from the downregulation of the amino acid transporter, thereby explaining the absence of any significant difference in BW between the experimental and control groups on day 7. The advantageous development of the intestinal villi consequently contributes to an increased ADG in the glycine-treated groups during the subsequent week. In this study, the continuous provision of glycine through drinking water from birth to day 7 may have contributed to an excessive intake of amino acids, potentially leading to the downregulation of certain genes associated with intestinal tight junction proteins, cellular resistance against oxidation, and amino acid transport. Subsequent research endeavors will focus on investigating the ideal dosage of glycine in drinking water and the optimal duration of its administration.

## 5. Conclusions

This study describes in detail the effects of glycine supplementation in drinking water during the first week of life on the growth performance and intestinal development of chicks. Glycine supplementation in the drinking water has been found to enhance the ADG of chicks from 7 to 14 days, and the effect of 1% glycine supplementation is significant. However, glycine supplementation had no significant effect on most organ indexes in either period. Glycine supplementation also affected the jejunum villus height and V/C ratio on both days 7 and 14, thereby suggesting improved intestinal absorption efficacy. Additionally, glycine supplementation can significantly affect the mRNA expression level of the *ZO-1*, *GCLM*, and *rBAT* genes in jejunum. In summary, glycine supplementation in the drinking water of chicks has the potential to enhance growth performance and intestinal absorption capacity, and it may have resulted in changes in the expression of a few genes related to the mucosal immune barrier, cellular capacity against oxidation, and the amino acid transporter in jejunum, which require more research in the future.

## Figures and Tables

**Figure 1 animals-13-03109-f001:**
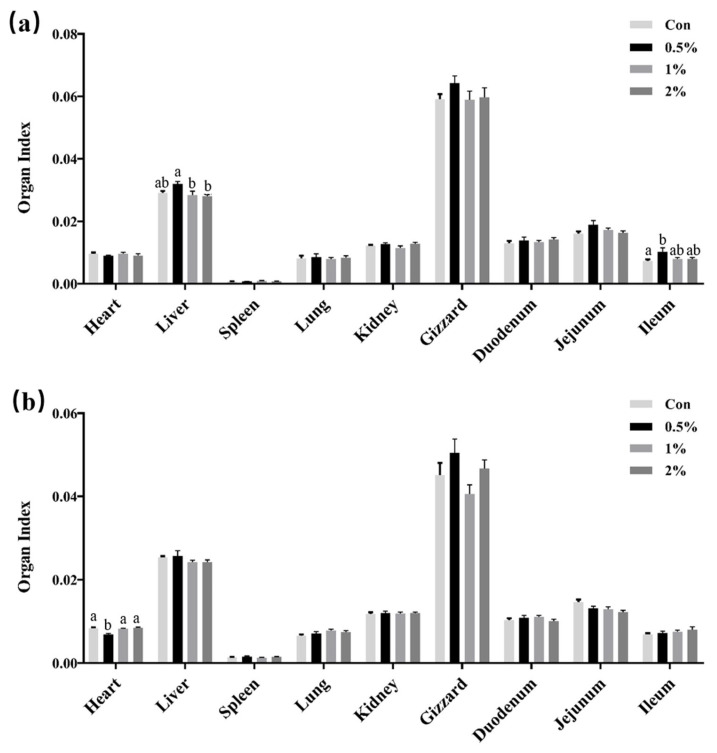
The effects of glycine supplementation in drinking water on organ index (absolute weight normalized to body weight) of chicks. (**a**) Organ index at day 7 and (**b**) organ index at day 14. The values were calculated using means with standard error of the mean. Data with different lower-case superscript letters are significantly different (*p* < 0.05).

**Figure 2 animals-13-03109-f002:**
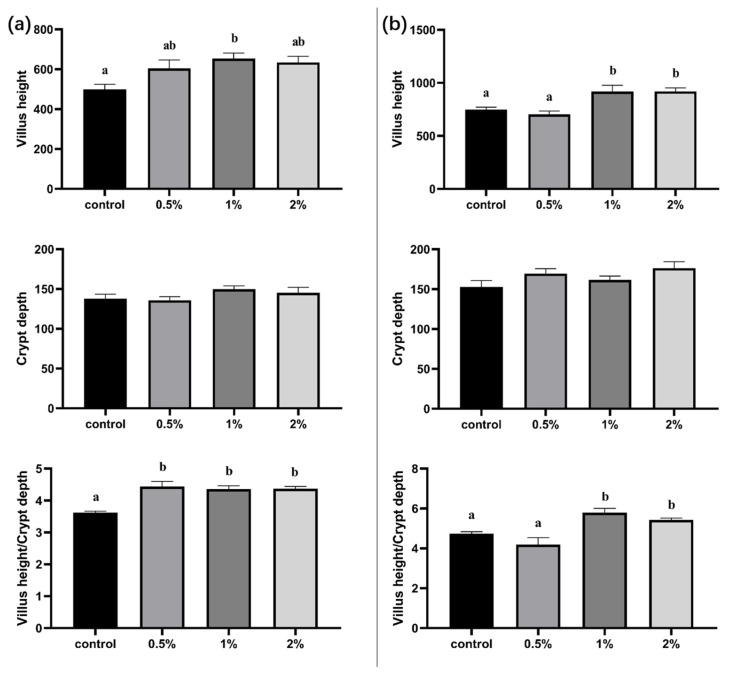
Effect of glycine supplementation in drinking water on villus height, crypt depth, and villus height/crypt depth of jejunum of chicks at day 7 (**a**) and day 14 (**b**). Different lowercase letters in the bar indicate a significant difference in different groups (*p* ≤ 0.05).

**Table 1 animals-13-03109-t001:** Primers used for quantitative RT-PCR.

Primer Names	Primer Sequence (5′→3′)	GenBank No
*Keap1*-F	GCCCTCAACAACTGCAT	MN416132.1
*Keap1*-R	CGGGTCGTAACACTCCA	
*NQO1*-F	TCAATGCCGTGCTCTCA	NM_001277620.2
*NQO1*-R	CAGCCGCTTCAATCTTC	
*GCLM*-F	TTCGGTCATTATTGCCC	NM_001007953.2
*GCLM*-R	ACCTGATTGCTGCTTGG	
*SOD1*-F	ATGTGACTGCAAAGGGAGGA	NM_205064.2
*SOD1*-R	AGCTAAACGAGGTCCAGCAT	
*GSR*-F	AGTGGCTTGCTGGAGGT	XM_040671422.1
*GSR*-R	GGGTCAGGAGGGCTTTG	
*GPX*-F	GACCAACCCGCAGTACATCA	NM_001277853.3
*GPX*-R	GAGGTGCGGGCTTTCCTTTA	
*GLYT1*-F	CGCCACTCTTCTTCCAG	NM_001031279.2
*GLYT1*-R	CTCCAGTAGCGTGTCCC	
*PepT1*-F	CTTGGCAGATCCCTCAGTATTT	XM_034074354.1
*PepT1*-R	GTTGGGCTTCAACCTCATTTG	
*B^0^AT1*-F	CATGATCGGACACAAGCCCA	XM_419056.6
*B^0^AT1*-R	AGCATAGACCCAGCCAGGATA	
*rBAT*-F	GGAGAGGCACGAAGTGAAAT	XM_010727793.3
*rBAT*-R	CGAGGGTAGACCTGGTAGATAG	
*Occludin*-F	AGCCCTCAATACCAGGATGTG	NM_205128.1
*Occludin*-R	CGCTTGATGTGGAAGAGCTTG	
*ZO-1*-F	AAGAGGAAGCTGTGGGTAACTC	XM_040680632.1
*ZO-1*-R	TGAAGAGTCACCGTGTGTTGT	
*ZO-2*-F	CCTACATTGGTTCAAGCATCGTGA	NM_001277622.1
*ZO⁃2*-R	GATGTCGGGAGGCAGGTTGA	
*Claudin1*-F	AGAAGATGCGGATGGCT	NM_001013611.2
*Claudin1*-R	AACGGGTGTGAAAGGGT	
*Claudin2*-F	GATACGTGTAGCAGCAGCAG	NM_001277622.1
*Claudin2*-R	AGCTGGGATTTCTGAGCAGT	
*Claudin3*-F	GTCTATGGGGCTGGAGATCG	NM_204202.2
*Claudin3*-R	ATCCACAGCCCTTCCCAGA	
*POLR2B*-F	TTACAAGCAGAAGCCCAGC	NM_001006448.2
*POLR2B*-R	TCACAAAGGTCACGGTCAGT	
*AP2M1*-F	AAAAACAGGGCAAAGGGACT	NM_001079494.2
*AP2M1*-R	CACGGAAGGGAAGGATGATG	

**Table 2 animals-13-03109-t002:** Effect of glycine supplementation in drinking water on body weight and average daily gain of chicks. ADG, average daily gain.

Items	Treatments				*p*-Value
	Control	0.5%	1%	2%	
Body weight, g					
0 d	38.80 ± 1.07	40.47 ± 0.70	39.05 ± 0.72	39.75 ± 0.84	0.51
7 d	91.59 ± 1.34	93.83 ± 2.03	93.34 ± 1.42	95.31 ± 1.80	0.48
14 d	164.12 ± 2.66	166.59 ± 4.87	174.00 ± 3.46	175.72 ± 4.16	0.46
ADG, g					
0–7 d	7.54 ± 0.09	7.62 ± 0.22	7.76 ± 0.12	7.94 ± 0.17	0.33
7–14 d	10.36 ± 0.18 ^a^	10.39 ± 0.41 ^ab^	11.52 ± 0.29 ^b^	11.49 ± 0.40 ^ab^	0.01

^a,b^ Means within a line with different superscripts are significantly different (*p* < 0.05).

**Table 3 animals-13-03109-t003:** Relative mRNA expression of tight junction protein-related genes in jejunum. Abbreviations: *ZO-1*, tight junction protein 1; *ZO-2*, tight junction protein 2.

Items	Treatments				*p*-Value
	Control	0.5%	1%	2%	
*Occludin*	1.00 ± 0.16	1.06 ± 0.15	0.90 ± 0.16	0.67 ± 0.13	0.41
*ZO-1*	1.00 ± 0.22 ^a^	2.12 ± 0.39 ^ab^	1.81 ± 0.25 ^ab^	2.55 ± 0.50 ^b^	<0.05
*ZO-2*	1.00 ± 0.21	0.52 ± 0.08	0.74 ± 0.07	0.59 ± 0.05	0.07
*Claudin1*	1.00 ± 0.18	1.03 ± 0.16	0.60 ± 0.09	0.62 ± 0.10	0.08
*Claudin2*	1.00 ± 0.15	1.22 ± 0.22	0.68 ± 0.06	0.87 ± 0.15	0.14
*Claudin3*	1.00 ± 0.55	0.33 ± 0.04	0.44 ± 0.08	0.56 ± 0.25	0.47

^a,b^ Mean within a line with different superscripts are significantly different (*p* < 0.05).

**Table 4 animals-13-03109-t004:** Relative mRNA expression of Keap1/Nrf2 signaling pathway-related genes in jejunum. Abbreviations: *Keap1*, Kelch-like ECH-associated protein 1; *NQO1*, NAD(P)H quinone dehydrogenase 1, *GCLM*, glutamate–cysteine ligase modifier subunit; *SOD1*, superoxide dismutase 1; *GSR*, glutathione–disulfide reductase; *GPX*, glutathione peroxidase.

Items	Treatments				*p*-Value
	Control	0.5%	1%	2%	
*Keap1*	1.00 ± 0.41	1.53 ± 0.25	1.92 ± 0.48	1.42 ± 0.28	0.40
*NQO1*	1.00 ± 0.17	1.72 ± 0.38	1.28 ± 0.16	0.84 ± 0.21	0.19
*GCLM*	1.00 ± 0.16 ^a^	0.59 ± 0.08 ^ab^	0.78 ± 0.17 ^ab^	0.41 ± 0.09 ^b^	0.04
*SOD1*	1.00 ± 0.12	0.78 ± 0.07	0.69 ± 0.06	0.89 ± 0.16	0.26
*GSR*	1.00 ± 0.24	0.71 ± 0.05	0.57 ± 0.11	0.58 ± 0.08	0.16
*GPX*	1.00 ± 0.04	0.84 ± 0.17	0.61 ± 0.05	0.77 ± 0.09	0.11

^a,b^ Mean within a line with different superscripts are significantly different (*p* < 0.05).

**Table 5 animals-13-03109-t005:** Relative mRNA expression of amino acid transporter genes in jejunum. Abbreviations: *GLYT1*, solute carrier family 6 member 9; *PepT1*, solute carrier family 15 member 1; *B^0^AT1*, solute carrier family 6 member 19; *rBAT*, solute carrier family 3 member 1.

Items	Treatments				*p*-Value
	Control	0.5%	1%	2%	
*GLYT1*	1.00 ± 0.18	1.90 ± 0.34	0.90 ± 0.15	0.67 ± 0.18	0.20
*PepT1*	1.00 ± 0.18	0.83 ± 0.18	0.76 ± 0.13	0.66 ± 0.14	0.52
*B^0^AT1*	1.00 ± 0.18	0.71 ± 0.12	0.78 ± 0.10	0.70 ± 0.06	0.34
*rBAT*	1.00 ± 0.04 ^a^	0.80 ± 0.10 ^a^	0.49 ± 0.06 ^b^	0.49 ± 0.08 ^b^	<0.01

^a,b^ Mean within a line with different superscripts are significantly different (*p* < 0.05).

## Data Availability

The data presented in this study are available on request from the authors.

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
