# Peer review of "Effects of Glycine Supplementation in Drinking Water on the Growth Performance, Intestinal Development, and Genes Expression in the Jejunum of Chicks"

_animals, 2023, doi:10.3390/ani13193109_

Round 1
Reviewer 1 Report
Manuscript is interesting and suggesting important insights regarding chicks' nutrition and intestine development.
Yet, it suffers from some major problems, which should and can be dealt with.
Simple summary and abstract:
· Well written
· Please add to the abstract the number of chicks used in this research.
Introduction:
· Please add a short paragraph regarding the relevant parameters that were investigated: transporters, anti-oxidants and tight junction genes.
Material and methods:
· In general, section should be revised to ensure proper scientific writing (mostly reagents and instruments manufacture data), which is not always present (4% formaldehyde solution, phosphate buffers, paraffin wax and etc.).
· Sub-section 2.2: were the chicks food deprived prior to euthanasia, to ensure empty GI? Were the GI organs emptied prior to weighing them? – Please add this data.
· Have you checked for glycine present in the potable water? If so, please add this data.
· Using only one housekeeping gene is problematic; it is acceptable to use at least 2. Please add an additional reference supporting the used housekeeping gene for this purpose.
Results:
· Lines 143- 144: referred data regarding BW is presented in table 2 (not 3) – please revise.
· Table 2: for each mean, there should be corresponded SEM. At the moment, a single SEM is presented without attribution to a specific group – this is an important data, thus this issue this must be revised. Accordingly, if variances (SEM) are not equal, you may need to change your statistical analysis method.
· Lines 157-161: regarding the liver index- it should be emphasize that statistically it is not different from control. Regarding the ileum index – it should be emphasize that it is only statistically different from the control group.
· Sub-section 3.3: results description regarding crypt depth is missing from the text – please revise.
· Table 3 and 6 are divided between 2 pages – please, make sure the entire table is located at the same page.
· Tables 4,5,6: same comment as to table 2; SEM should be presented for each group.
· In each table, the authors describe only statistically changed protein - other proteins results description is missing – this issue must be resolved.
Discussion and conclusions:
· Line 226: please remove extra gap after the word chicks,
· Lines 283-285: the sentence is not clear and repeats the said in line 280 – please revise.
· Lines 296-303: appears last in the discussion but refers to data presented second in line – please keep the same order in the discussion and relocate this paragraph post line 257.
· In my opinion, this sections suffers from 2 main faults:
o Authors should be accurate and avoid sentences like "our study revealed alterations in the expression of 288 amino acid transport genes", when according to the presented results, only one gene expression was changed.
o There is a problem of scientific soundness and still a lot of gaps to fill such as: why only 1/6 investigated tight junction genes was altered? And regarding the antioxidants genes (1/6) and the transporters genes (1/4)? What is the rational in the observation in which antioxidant related gene is down regulated? The chicks were not in heat stress conditions. What is the rational for changes observed at 7-14 days, in a timeline without the supplementation of glycine? In several parameters, 1% glycine supplementation is better than 0.5%, but also than 2%; what is the rational for that?
o This entire section should be revised to solve as much gaps as possible.
Minor english editing is required
Author Response
Simple summary and abstract:
Well written
Please add to the abstract the number of chicks used in this research.
A: Thanks for your suggestions. We have added the number of chicks to the abstract
Introduction:
Please add a short paragraph regarding the relevant parameters that were investigated: transporters, anti-oxidants and tight junction genes.
A: We have added the relevant parameters of transporters, anti-oxidants and tight junction genes to the Introduction, see line 83-88.
Material and methods:
In general, section should be revised to ensure proper scientific writing (mostly reagents and instruments manufacture data), which is not always present (4% formaldehyde solution, phosphate buffers, paraffin wax and etc.).
A: Thanks for your suggestions. We have added relevant data to the manuscript.
Sub-section 2.2: were the chicks food deprived prior to euthanasia, to ensure empty GI? Were the GI organs emptied prior to weighing them? – Please add this data.
A: The chickens fasted for 12 hours before euthanasia, and the GI were emptied prior to weigh them. We have also added relevant descriptions to the manuscript (line 113-115).
Have you checked for glycine present in the potable water? If so, please add this data.
A: We didn’t test the glycine in drinking water. So, sorry about that, this data can’t be added in the manuscript.
Using only one housekeeping gene is problematic; it is acceptable to use at least 2. Please add an additional reference supporting the used housekeeping gene for this purpose.
A: Thanks for your suggestion. We re-performed the qRT-PCR with two housekeeping genes and re-analyzed the data.
Results:
Lines 143- 144: referred data regarding BW is presented in table 2 (not 3) – please revise.
A: Thanks for your suggestion. We have revised it in the manuscript.
Table 2: for each mean, there should be corresponded SEM. At the moment, a single SEM is presented without attribution to a specific group – this is an important data, thus this issue this must be revised. Accordingly, if variances (SEM) are not equal, you may need to change your statistical analysis method.
A: Thanks for your suggestion. Our table shows a pooled SEM of four groups, not attribution to a specific group. Therefore, it can reflect the overall data variation. This is a common representation, which is the same as the statistical method in previous studies [1-3]. And the Animals journal agrees with this way of presenting the data [1]. If the reviewer insists on revising it, we will modify it again.
[1] Wu D, Zhang Z, Shao K, Wang X, Huang F, Qi J, Duan Y, Jia Y, Xu M. Effects of Sodium Butyrate Supplementation in Milk on the Growth Performance and Intestinal Microbiota of Preweaning Holstein Calves[J]. Animals, 2023,13(13).
[2] Deng C, Zheng J, Zhou H, You J, Li G. Dietary glycine supplementation prevents heat stress-induced impairment of antioxidant status and intestinal barrier function in broilers[J]. Poult Sci, 2023,102(3):102408.
[3] Yu L L, Gao T, Zhao M M, Lv P A, Zhang L, Li J L, Jiang Y, Gao F, Zhou G H. In ovo feeding of L-arginine alters energy metabolism in post-hatch broilers[J]. Poult Sci, 2018,97(1):140-148.
Lines 157-161: regarding the liver index- it should be emphasize that statistically it is not different from control. Regarding the ileum index – it should be emphasize that it is only statistically different from the control group.
A: Thanks for your suggestion. We have made revision to this section (line 173-176).
Sub-section 3.3: results description regarding crypt depth is missing from the text – please revise.
A: We have added a description of crypt depth to the manuscript (line 190).
Table 3 and 6 are divided between 2 pages – please, make sure the entire table is located at the same page.
A: We have made sure that the entire table is on the same page.
Tables 4,5,6: same comment as to table 2; SEM should be presented for each group.
A: Thanks for your suggestion. Our table shows a pooled SEM of four groups, not attribution to a specific group. Therefore, it can reflect the overall data variation. This is a common representation, which is the same as the statistical method in previous studies. And the Animals journal agrees with this way of presenting the data. If the reviewer insists on revising it, we will modify it again.
In each table, the authors describe only statistically changed protein - other proteins results description is missing – this issue must be resolved.
A: Thanks for your suggestion. We have added the above information to the manuscript as suggested.
Discussion and conclusions:
Line 226: please remove extra gap after the word chicks.
A: We have deleted the gap.
Lines 283-285: the sentence is not clear and repeats the said in line 280 – please revise.
A: Thanks for your suggestion. We have made revision to this section (line 312).
Lines 296-303: appears last in the discussion but refers to data presented second in line – please keep the same order in the discussion and relocate this paragraph post line 257.
A: We rearranged the discussion sections according to the order of the results.
In my opinion, this sections suffers from 2 main faults:
Authors should be accurate and avoid sentences like "our study revealed alterations in the expression of 288 amino acid transport genes", when according to the presented results, only one gene expression was changed.
A: We have checked the full text to make the expression of the sentences clearer.
There is a problem of scientific soundness and still a lot of gaps to fill such as: why only 1/6 investigated tight junction genes was altered? And regarding the antioxidants genes (1/6) and the transporters genes (1/4)? What is the rational in the observation in which antioxidant related gene is down regulated? The chicks were not in heat stress conditions. What is the rational for changes observed at 7-14 days, in a timeline without the supplementation of glycine? In several parameters, 1% glycine supplementation is better than 0.5%, but also than 2%; what is the rational for that?
A: Although glycine supplementation only significantly affected the expression changes of 3 genes in the identified genes, the results showed that the expression levels of few other genes were also changed, although with no statistical significance. The effects of glycine supplementation on the expression of genes related to tight junction, antioxidants and transporters in the intestine are different in different species, but basically affect the expression of one or a few genes, which may be attributed to interspecies variations.
The chicks were not in heat stress conditions. However, we believe that the chicks experienced the process of internal nutrition to external nutrition in the first week, which is prone to intestinal oxidative stress. The glycine supplementation alleviates oxidative stress, which decreases the expression of antioxidant related gene.
Without the supplementation of glycine, the ADG from 7 to14 days, the villus height and V/C at 14 days of the group were still improved. The improvement of ADG may due to that the glycine supplementation at 0-7 days can promote intestinal villi development and absorption capacity, and this advantage effect is continuous. The increase in villus height and V/C may depend on a better early developmental foundation and glycine storage.
In several parameters, 1% glycine supplementation is better than 2%. We think it may be because high concentration of glycine supplement enhance white-fat loss and improve sensitivity to insulin in animals. At the same time, reasonable dose selection is the necessary condition for the protective effect of glycine on cells, and too high concentration of glycine has toxic effect on cells. Therefore, it is not the higher the concentration of glycine, the better the effect, 1% of glycine concentration may be more reasonable
The above comments have also discussed in the Discussion section.
This entire section should be revised to solve as much gaps as possible.
A: We have addressed the corresponding problems and filled in the gaps according to the comments.
Reviewer 2 Report
This paper investigates the effects of glycine on chicks. The manuscript describes how glycine can enhance growth performance and promote intestinal health in young chickens. The article further explores the relationship between glycine and intestinal absorption capacity as well as gut health. However, a few areas of this paper need attention. I recommend this for publication after the authors have addressed the following.
1. Simple Summary must contain most of the key information of the paper in brief form.
2. Line 89,Survival rates were not addressed.
3. Line 95,Whether ten birds are expressed appropriately?
4. Line 102,How to choose the glycine concentration?
5. Table 1,Why mRNA expression of these genes is done?
6. Line125,Whether the conditions for Quantitative Real-Time PCR experiments can be supplemented?
7. Gene expression does not indicate protein expression and can provide protein expression data?
Extensive editing of English language required.
Author Response
This paper investigates the effects of glycine on chicks. The manuscript describes how glycine can enhance growth performance and promote intestinal health in young chickens. The article further explores the relationship between glycine and intestinal absorption capacity as well as gut health. However, a few areas of this paper need attention. I recommend this for publication after the authors have addressed the following.
- Simple Summary must contain most of the key information of the paper in brief form.
A: We have amended the Simple Summary section to include the key information of the paper in brief form.
- Line 89,Survival rates were not addressed.
A: The meaning of this sentence is that the healthy development of chicks can improve the survival rate. In order to avoid misunderstanding, we have revised this sentence.
- Line 95,Whether ten birds are expressed appropriately?
A: We have modified ten birds to ten chicks.
- Line 102,How to choose the glycine concentration?
A: Through reading reference, we found that the concentration of glycine added through diet or drinking water is usually 0.5%, 1% and 2%, too low concentration is ineffective, and too high concentration may cause toxicity. Therefore, according to the experience of previous studies, we chose these concentration.
[1] Deng C, Zheng J, Zhou H, You J, Li G. Dietary glycine supplementation prevents heat stress-induced impairment of antioxidant status and intestinal barrier function in broilers[J]. Poult Sci, 2023,102(3):102408.
[2] Fan X, Li S, Wu Z, Dai Z, Li J, Wang X, Wu G. Glycine supplementation to breast-fed piglets attenuates post-weaning jejunal epithelial apoptosis: a functional role of CHOP signaling[J]. Amino Acids, 2019,51(3):463-473.
- Table 1,Why mRNA expression of these genes is done?
A: We have found through previous studies that in other animals and chickens of other ages, glycine supplementation enhances antioxidant capacity and barrier function development of intestinal tract and may affect amino acid transport. Therefore, we selected the genes in Table 1 to identify their mRNA expression levels. The relevant introduction has been added to line 83-88.
- Line125,Whether the conditions for Quantitative Real-Time PCR experiments can be supplemented?
A: We have supplemented the conditions for Quantitative Real-Time PCR experiments (line 139-143).
- Gene expression does not indicate protein expression and can provide protein expression data?
A: Thanks for your advice. It's hard to find the suitable antibodies in chickens. We tried to detect protein levels of related genes using antibodies from different companies, but we did not find suitable antibodies. Therefore, the test results are not available.
Reviewer 3 Report
The authors want to test the dose effect glycine supplementation in drinking water on growth parameters, intestinal development and gene expression in jejunum in newly-born chickens.
We present some suggestions for change, inclusion or correction that we consider relevant to clarify the work considering the fulfillment of its objectives.
GENERAL COMMENT
Define the acronym for the first time the term or expression is used. Thereafter, use only the acronym.
I think that the word “potable” should be removed from the entire work because it seems redundant, in relation to the mandatory hygiene, physical and microbiological conditions that water for animal consumption, particularly for chickens, must fulfill. Perhaps using “drinking water”.
TITLE
The title should be adjusted to reflect the aims of this study. It should include mention of genes expression in the jejunum or, indirectly, the antioxidant effect and amino acid transport of the genes expression studied.
ABSTRACT
The abstract reflects the sequence of the work, principal results, and conclusions.
INTRODUCTION
Read more carefully and rephrase some sentences for better understanding the state of the art.
The sentence from lines 56 to 58 is not very clear because symbiotic bacteria have a positive relationship with the host, except in some cases when their proliferation is exacerbated (e.g. E. coli).
Define the aims of the study more concretely and objectively (lines 81 to 89).
MATERIAL AND METHODS
Clarify the term basal vs base. Is not the same thing. I think you might be referring to the base diet (line 96).
In section 2.2. you should also mention the measurement of live weight. You don't mention food consumption (except in the discussion-Line 233). Has it been evaluated? It would be interesting if it could be mentioned explicitly to better clarify the efficiency of increasing BW and ADG with supplementation of glycine.
In Section 2.3 you should include the degree of magnification of the microscope lens.
The sentence on lines 132-133 should be included in Section 2.5 (Statistical Analysis).
RESULTS
Consider review some of the titles of the tables and figures to better discriminate the data presented. And delete the final part of the title (abbreviations or notes) because they are also mentioned at the footnote of tables or include them at the footnote of tables when they don't exist there.
The section 3.1 must be renamed as “Body weight and average daily gain”.
In line 143 the correct mention is for table 2, not 3.
In line 175, after the first sentence you must do reference to the table 3 between brackets.
In lines 177-178 the sentence is not true as the higher value relates to 0.5% supplementation.
At the end of section 3.3 it should be noted that there are no significant differences between groups for crypt depth.
In table 3, remove the superscripts for columns 7d and 14d from the crypth depth as there are no significant differences in both cases.
At the footnote of table 3 is not Raito but Ratio.
The section 3.4 must be renamed; include just the “Gene Expression in Jejunum”. Make more sense!
DISCUSSION
The sequential description of the results discussion is a bit difficult to understand, and it is not entirely clear. Readjust and clarify.
The sentence in lines 225-227 should be clarified and revised.
CONCLUSION
Review and clarify, including conclusions on all aspects considered: zootechnical performance, morphology and integrity of the intestine, relative weight of organs, expression of genes in the intestine and putative protective/disruptive effects.
I think there is some lack of care in the overall writing of the text. Review carefully the grammar, syntax and vocabulary and adjust some small formatting aspects, making the text clearer and fluid. Some words do not seem to me to be the most appropriate in this type of technical text, breaking the fluidity of the most appropriate syntax, despite the same meaning, although somewhat mismatched.
Author Response
GENERAL COMMENT
Define the acronym for the first time the term or expression is used. Thereafter, use only the acronym.
A: Thanks for your suggestion. We have reviewed and revised them in the manuscript.
I think that the word “potable” should be removed from the entire work because it seems redundant, in relation to the mandatory hygiene, physical and microbiological conditions that water for animal consumption, particularly for chickens, must fulfill. Perhaps using “drinking water”.
A: Thanks for your suggestion. We have modified "potable water" to "drinking water".
TITLE
The title should be adjusted to reflect the aims of this study. It should include mention of genes expression in the jejunum or, indirectly, the antioxidant effect and amino acid transport of the genes expression studied.
A: We have modified the TITLE to "Effects of Glycine Supplementation in Drinking Water on the Growth Performance, Intestinal Development, and Genes Expression in the Jejunum of chicks”
ABSTRACT
The abstract reflects the sequence of the work, principal results, and conclusions.
A: We have improved the ABSTRACT section.
INTRODUCTION
Read more carefully and rephrase some sentences for better understanding the state of the art.
A: Thank you for your suggestion. We have invited English language professionals to revise the manuscript.
The sentence from lines 56 to 58 is not very clear because symbiotic bacteria have a positive relationship with the host, except in some cases when their proliferation is exacerbated (e.g. E. coli).
A: In the case of a healthy intestinal tract, symbiotic bacteria have a positive relationship with the host. However, when the intestinal barrier is damaged, pathogenic bacteria and symbiotic bacteria will invade the host intestinal epithelium.
We have revised this sentence to avoid ambiguity (line 59-60).
Define the aims of the study more concretely and objectively (lines 81 to 89).
A: We have defined the aims of the study more concretely and objectively (line 89-94).
MATERIAL AND METHODS
Clarify the term basal vs base. Is not the same thing. I think you might be referring to the base diet (line 96).
A: Thank you for your suggestion. We have modified "basal diet" to "base diet".
In section 2.2. you should also mention the measurement of live weight. You don't mention food consumption (except in the discussion-Line 233). Has it been evaluated? It would be interesting if it could be mentioned explicitly to better clarify the efficiency of increasing BW and ADG with supplementation of glycine.
A: We have added the measurement of live weight (line 114-115).
We did not collect statistics on food consumption, which will be taken into account in subsequent studies.
In Section 2.3 you should include the degree of magnification of the microscope lens.
A: We have added the degree of magnification of the microscope.
The sentence on lines 132-133 should be included in Section 2.5 (Statistical Analysis).
A: We have included the analysis method of qRT-PCR results in Section 2.5.
RESULTS
Consider review some of the titles of the tables and figures to better discriminate the data presented. And delete the final part of the title (abbreviations or notes) because they are also mentioned at the footnote of tables or include them at the footnote of tables when they don't exist there.
A: Thank you for your suggestions. We have revised the titles of the tables and figures, and made sure that abbreviations or notes only appear in the title or footnote.
The section 3.1 must be renamed as “Body weight and average daily gain”.
A: The subheading of section 3.1 has been revised to “Body weight and average daily gain”.
In line 143 the correct mention is for table 2, not 3.
A: We have revised this mistake.
In line 175, after the first sentence you must do reference to the table 3 between brackets.
A: We have added a reference to table 3.
In lines 177-178 the sentence is not true as the higher value relates to 0.5% supplementation.
A: We have revised this sentence to make it more accurate.
At the end of section 3.3 it should be noted that there are no significant differences between groups for crypt depth.
A: We have added a description of the crypt depth.
In table 3, remove the superscripts for columns 7d and 14d from the crypt depth as there are no significant differences in both cases.
A: We have removed the superscripts.
At the footnote of table 3 is not Raito but Ratio.
A: We have revised this mistake.
The section 3.4 must be renamed; include just the “Gene Expression in Jejunum”. Make more sense!
A: We have revised the subheading of section 3.4.
DISCUSSION
The sequential description of the results discussion is a bit difficult to understand, and it is not entirely clear. Readjust and clarify.
A: We have adjusted the order of the discussion sections to make its contents clearer.
The sentence in lines 225-227 should be clarified and revised.
A: We have revised the sentence to make it clearer.
CONCLUSION
Review and clarify, including conclusions on all aspects considered: zootechnical performance, morphology and integrity of the intestine, relative weight of organs, expression of genes in the intestine and putative protective/disruptive effects.
A: We have reviewed and clarified the CONCLUSION section to include the conclusions of all aspects considered.
Comments on the Quality of English Language
I think there is some lack of care in the overall writing of the text. Review carefully the grammar, syntax and vocabulary and adjust some small formatting aspects, making the text clearer and fluid. Some words do not seem to me to be the most appropriate in this type of technical text, breaking the fluidity of the most appropriate syntax, despite the same meaning, although somewhat mismatched.
A: Thank you for your suggestion. We have carried out a careful review of the full text and invited English language professionals to revise the article.
Round 2
Reviewer 1 Report
The revised version is much better, yet two major issues still present and require mending: 1. Adequate results presentation, 2. The discussion section still consist lots of gaps, lacks scientific soundness and integrative interpretation. This issues must be fixed.
Yet, it suffers from some major problems, which should and can be dealt with.
Simple summary and abstract:
· Well written
Introduction:
· Well written
Material and methods:
· In general, section should be revised to ensure proper scientific writing (mostly reagents and instruments manufacture data), which is not always present (4% formaldehyde solution, phosphate buffers, paraffin wax and etc.).
· A: Thanks for your suggestions. We have added relevant data to the manuscript
· R2: this problem was partially solved. It is acceptable to write the name of the manufacture along with the country of production – please add the missing data.
Results:
· Table 2: for each mean, there should be corresponded SEM. At the moment, a single SEM is presented without attribution to a specific group – this is an important data, thus this issue this must be revised. Accordingly, if variances (SEM) are not equal, you may need to change your statistical analysis method.
· A: Thanks for your suggestion. Our table shows a pooled SEM of four groups, not attribution to a specific group. Therefore, it can reflect the overall data variation. This is a common representation, which is the same as the statistical method in previous studies [1-3]. And the Animals journal agrees with this way of presenting the data [1]. If the reviewer insists on revising it, we will modify it again.
· R2: I don’t understand the logic of providing a single SEM for a pool mean from different treatment. In order to accurately assess the variability between groups and provide a correct statistical analysis – you should provide a SEM for each treatment group following validation that the suitable statistical model was used – please revise according to the initial comment.
· Line 175: please add the word "only" after the word was.
· Table 4 is divided between 2 pages – please, make sure the entire table is located at the same page.
· Tables 4,5,6: same comment as to table 2; SEM should be presented for each group.
· A: Thanks for your suggestion. Our table shows a pooled SEM of four groups, not attribution to a specific group. Therefore, it can reflect the overall data variation. This is a common representation, which is the same as the statistical method in previous studies. And the Animals journal agrees with this way of presenting the data. If the reviewer insists on revising it, we will modify it again. In each table, the authors describe only statistically changed protein - other proteins results description is missing – this issue must be resolved.
· R2: same comment as for table 2.
Discussion and conclusions:
· Line 273: please remove extra gap after the words "on" and "the".
· Lines 247-248: the authors claim "However, the glycine content 247 in the base diet merely satisfies metabolic requirements of the chicks" – please add reference to support this claim.
· Lines 248-250: it is not acceptable to add recommendations on the beginning of the discussion section, prior to the current results interpretation. Please, relocate these sentences to the end of the conclusions section.
· Line 296: the word "salmonella" should be in italics – please revise.
· Lines 288-307: the entire section gives a good background to the importance of tight junction proteins and their relevance to glycine. But, the part you discuss your current results is missing. – Please revise.
· Lines 308-319:
o Similarly to the previous comment, you discuss only GCLM gene without addressing all other anti-oxidative genes (as you mention in the results section, some increase and some decrease, even if not significant) – you should revise this sub-section to give the reader holistic and integrative explanation to your results.
o Still, the main focus in the interpretation to the GCLM is base on observations seen in heat stressed chicken – this is not the case in your current research. – Please revise.
o The authors claim "We believe that the chicks experienced the process of internal nutrition to external nutrition in the first week, which is prone to intestinal oxidative stress" - please add reference/s to support this claims (regarding both the process and the oxidative stress).
o The authors claim "However, the glycine supplementation alleviates oxidative stress, which decreases the expression of antioxidant related genes" – but, as you mention in the results section, some of them increase – this explanation lacks scientific soundness and must be revised.
· Lines 319-325:
o Same comment as previous one, you only addressed rBAT gene out of 4 investigated genes. You should provide a holistic and integrative explanation.
o The authors claim "These findings may be attributed to interspecies variations" – I can accept that. But, is the functional/physiological explanation for this observation? – This is an important gap that must be closed.
· Line 333: the authors claim "thereby improving intestinal absorption efficiency" – this is a very conclusive sentence without data to back it up. You have not checked for intestinal absorption efficiency. There by, I suggest you should rephrase: "thereby, suggesting improved intestinal absorption efficacy".
· No problems
Author Response
The revised version is much better, yet two major issues still present and require mending: 1. Adequate results presentation, 2. The discussion section still consist lots of gaps, lacks scientific soundness and integrative interpretation. This issues must be fixed.
Yet, it suffers from some major problems, which should and can be dealt with.
A: Thank you for your suggestions. In this edition, we have provided sufficient result data and conducted a comprehensive discussion on the results such as qRT-PCR in the discussion section.
Simple summary and abstract:
Well written
Introduction:
Well written
Material and methods:
In general, section should be revised to ensure proper scientific writing (mostly reagents and instruments manufacture data), which is not always present (4% formaldehyde solution, phosphate buffers, paraffin wax and etc.).
A: Thanks for your suggestions. We have added relevant data to the manuscript
R2: this problem was partially solved. It is acceptable to write the name of the manufacture along with the country of production – please add the missing data.
A2: Thanks for your suggestions. We have supplemented the countries of manufacturers for the first time appeared in the manuscript.
Results:
Table 2: for each mean, there should be corresponded SEM. At the moment, a single SEM is presented without attribution to a specific group – this is an important data, thus this issue this must be revised. Accordingly, if variances (SEM) are not equal, you may need to change your statistical analysis method.
A: Thanks for your suggestion. Our table shows a pooled SEM of four groups, not attribution to a specific group. Therefore, it can reflect the overall data variation. This is a common representation, which is the same as the statistical method in previous studies [1-3]. And the Animals journal agrees with this way of presenting the data [1]. If the reviewer insists on revising it, we will modify it again.
R2: I don’t understand the logic of providing a single SEM for a pool mean from different treatment. In order to accurately assess the variability between groups and provide a correct statistical analysis – you should provide a SEM for each treatment group following validation that the suitable statistical model was used – please revise according to the initial comment.
A2: Thanks for your suggestion. We have added SEM for each treatment group in the Table 2. Considering that after completing the SEM in Table 3, the table is too long to be fully displayed in the document, we have presented the results in the form of Figure 2. Now, Table 3 has been changed to Figure 2, and the order of other Tables has also changed accordingly.
Line 175: please add the word "only" after the word was.
A: Thanks for your suggestion, we have added the “only” word.
Table 4 is divided between 2 pages – please, make sure the entire table is located at the same page.
A: Thanks for your comment. We have made sure that tables is on the same page. Table 4 is divided into 2 pages probably because we use different versions of Office software. Please view the PDF version of the manuscript we uploaded, Table 4 is on the same page.
Tables 4,5,6: same comment as to table 2; SEM should be presented for each group.
A: Thanks for your suggestion. Our table shows a pooled SEM of four groups, not attribution to a specific group. Therefore, it can reflect the overall data variation. This is a common representation, which is the same as the statistical method in previous studies. And the Animals journal agrees with this way of presenting the data. If the reviewer insists on revising it, we will modify it again.
R2: same comment as for table 2.
A2: Thanks for your comment. We have added SEM data in each group of Tables 3, 4, and 5. The original tables 4, 5, and 6 has been changed to Tables 3, 4, and 5, because original Table 3 has been changed to Figure 2, the subsequent table numbers have been changed accordingly.
Discussion and conclusions:
Line 273: please remove extra gap after the words "on" and "the".
A: Thanks for your suggestion. We have delated the extra gap.
Lines 247-248: the authors claim "However, the glycine content 247 in the base diet merely satisfies metabolic requirements of the chicks" – please add reference to support this claim.
A: Thanks for your suggestion. We are sorry that we did not find a suitable reference. This sentence is a subjective summary after reading some papers. We have deleted this sentence and restated our experimental purpose.
Lines 248-250: it is not acceptable to add recommendations on the beginning of the discussion section, prior to the current results interpretation. Please, relocate these sentences to the end of the conclusions section.
A: Thanks for your suggestion. We have revised these sentences.
Line 296: the word "salmonella" should be in italics – please revise.
A: Thanks for your suggestion. We have revised this issue.
Lines 288-307: the entire section gives a good background to the importance of tight junction proteins and their relevance to glycine. But, the part you discuss your current results is missing. – Please revise.
A: Thanks for your suggestion. We have added the relevant discussion to the current results.
Lines 308-319:
Similarly to the previous comment, you discuss only GCLM gene without addressing all other anti-oxidative genes (as you mention in the results section, some increase and some decrease, even if not significant) – you should revise this sub-section to give the reader holistic and integrative explanation to your results.
A: Thanks for your suggestion. We have added the discussion of all the antioxidant genes.
Still, the main focus in the interpretation to the GCLM is base on observations seen in heat stressed chicken – this is not the case in your current research. – Please revise.
A: Thanks for your suggestion. We have deleted this sentence and provided an explanation for the possible reasons for the downregulation of GCLM in the glycine group.
The authors claim "We believe that the chicks experienced the process of internal nutrition to external nutrition in the first week, which is prone to intestinal oxidative stress" - please add reference/s to support this claims (regarding both the process and the oxidative stress).
A: Thanks for your suggestion, we have deleted this sentence and discussed the causes of changes in the expression of antioxidative stress genes from other perspectives.
The authors claim "However, the glycine supplementation alleviates oxidative stress, which decreases the expression of antioxidant related genes" – but, as you mention in the results section, some of them increase – this explanation lacks scientific soundness and must be revised.
A: Thanks for your suggestion. We have discussed both the increased and decreased expression of antioxidant stress genes.
Lines 319-325:
Same comment as previous one, you only addressed rBAT gene out of 4 investigated genes. You should provide a holistic and integrative explanation.
A: Thanks for your suggestion. We have added discussions on other protein transporter genes and discussed possible reasons for differential expression.
The authors claim "These findings may be attributed to interspecies variations" – I can accept that. But, is the functional/physiological explanation for this observation? – This is an important gap that must be closed.
A: Thanks for your suggestion. We have deleted this sentence and provided an explanation for the possible reasons for the differences in amino acid transporters in the glycine group.
Line 333: the authors claim "thereby improving intestinal absorption efficiency" –this is a very conclusive sentence without data to back it up. You have not checked for intestinal absorption efficiency. There by, I suggest you should rephrase: "thereby, suggesting improved intestinal absorption efficacy".
A: Thanks for your suggestion, we have rephrased the sentence to "thereby, suggesting improved intestinal absorption efficacy" according to your suggestion.
